# Evaluation of Enamel Surfaces after Different Techniques of Interproximal Enamel Reduction

**DOI:** 10.3390/jfb14020110

**Published:** 2023-02-16

**Authors:** Francesca Silvestrini Biavati, Viola Schiaffino, Antonio Signore, Nicola De Angelis, Valentina Lanteri, Alessandro Ugolini

**Affiliations:** 1Department of Surgical and Integrated Diagnostic Sciences, University of Genova, 16132 Genoa, Italy; 2Therapeutic Dentistry Department, Institute of Dentistry, I.M. Sechenov First Moscow State Medical University, Trubetskaya Str., 119992 Moscow, Russia; 3Dental Department, University Trisakti, Jakarta 11440, Indonesia; 4Department of Biomedical Surgical and Dental Sciences, University of Milan, 20142 Milan, Italy

**Keywords:** enamel roughness, interproximal enamel reduction, stripping, orthodontics, crowding

## Abstract

According to the literature, interproximal enamel reduction (IER) has become a consolidated technique used in orthodontic treatments to gain space in particular situations such as dental crowding, non-extractive therapies, tooth-size discrepancies, and prevention of dental relapse. There are different methods to realize stripping, and enamel surfaces resulting after this procedure can be analyzed with SEM. The aim of this study was to analyze how different devices of IER leave the surface of the teeth. One hundred and sixty freshly extracted, intact human lower incisors were included in the study, fixed in a plaster support, and then processed with four different techniques of enamel reduction and finishing. Then, they were divided randomly into eight groups (A1–A2, B1–B2, C1–C2, D, and E), each containing twenty teeth. The A, B, and C groups were divided into two subgroups and then all the teeth were observed at SEM. Each digital image acquired by SEM showed that there were streaks on the surfaces, due to the cutter used. The results of this study showed that only group C2 (tungsten carbide bur followed by twelve steps of medium–fine–ultrafine 3M Soft Lex disks) has a few line, which is very similar to group E (untreated group), while the other groups have a lot of lines and show a rougher final surface.

## 1. Introduction

Orthodontic treatment represents a very difficult challenge for the clinicians because it aims to reach a perfect harmony between dental arches, cranial structures, and facial esthetic, in order to remain stable in time. Frequently, patients have big teeth and a lack of space, so it become difficult to correctly align the dental elements; this condition is often placed in the anterior region of dental arches and is an indicator of dental crowding [1].

Dental crowding and lack of space are two of the most frequent anomalies in orthodontics. About 50% of the untreated population between 15 and 50 years have mandibular crowding, from mild to moderate, and incisor irregularities [2].

There are many different techniques for gaining space in orthodontics. Extraction therapy is a decision that in some clinical situations represents the solution of choice, otherwise the expansion of the upper maxilla is one of the most used methods to avoid the extraction of healthy teeth [3,4,5]. Clinicians can also rely on distalization of the posterior teeth or proinclination of the anterior teeth, according to the clinical needs and the starting situation of the patient. In recent years, a growing number of orthodontic specialists have focused their interests on non-extraction therapy, and among the different options previously citated, interproximal enamel reduction (IER) has been widely accepted by researchers [6]. Interproximal enamel reduction is also called interdental stripping, enamel reproximation, and slenderizing [1,7,8], and it is a commonly applied technique in orthodontic treatment to obtain more space to align the incisors and maintain alignment in the long term [7,9].

IER is realized by removing no more than 50% of the initial enamel thickness from the mesial and distal surfaces of the teeth or of restorations, and then refining the resulting areas to obtain smooth and clean surfaces. The amount of enamel excised depends on the mesio-distal diameters of the teeth involved in the procedure. Each tooth has an individual anatomical variation, so the amount of interproximal enamel reduction changes from time to time and the space that can be reached is not always the same [2].

Generally, this procedure is more common on the lower incisors, because these teeth often have a triangular shape, with a mesio-distal/faciolingual index that is favorable to make stripping [10], but sometimes it is also necessary to be performed in the posterior sectors of the mouth.

When teeth are very crowded or they rotate, and the interproximal contact point is hidden, the use of separating elastics before the enamel reduction can help to provide the necessary space for the stripping procedure, especially for posterior teeth, also to avoid damaging the intact surfaces of adjacent teeth.

Literature on enamel reduction is varied, and consequently the entity of millimeters that can be removed is different depending on different authors. According to Sheridan [11], clinicians can obtain 2.5 mm of space by enamel reduction from five anterior contacts, and 6.4 mm when considering eight buccal contacts.

This procedure therefore offers an alternative to extraction therapy, because it significantly reduces treatment time and allows the transverse arch dimension and anterior inclination of inferior incisors to be maintained. The main indications for the IER procedure are the elimination of tooth-size discrepancies between the upper and lower arch (Bolton Index), the treatment of mild or moderate crowding, the normalization of gingival contour, the correction of the curve of Spee, the prevention of relapse with the enhancement of stability at the end of the orthodontic treatment, and the reshaping of the proximal contacts with the removal of “black triangles” due to interdental gingival retraction. IER can also be useful when the clinician has to reshape teeth, such as in congenitally missing lateral incisors when spaces are closed and canines have to be transformed in lateral incisors. Sometimes, this represents a useful procedure also in mixed dentition, when growing patients show a moderate crowding, or during functional therapy to redistribute spaces for permanent teeth that are yet to erupt [8].

This therapeutic procedure has been progressively improved in recent years, testing different methods in order to identify the most effective, efficient, and appropriate protocol that allows stripping to be performed by gaining the necessary space for the therapeutic outcomes and at the same time, reaching perfectly smooth surfaces of the tooth.

In an in vitro study, clinicians used a scanning electron microscopy (SEM) to evaluate the morphologic effect of a three-step technique, using an oscillating perforated diamond-coated disk for enamel reduction followed by a polishing procedure using 2 Soft-Lex XT disks. The SEM investigations demonstrated that more than 90% of the interproximal surfaces were very well-polished, resulting in polished enamel surfaces that were smoother than untreated enamel [7].

Although interdental enamel reduction has well-documented beneficial outcomes in orthodontics, some negative effects on the enamel surface have created debates [8].

It has been defined that 0.3–0.4 mm of enamel can be safely removed without rendering the enamel prone to dissolution, but this procedure modifies the ultra-morphological structures of the enamel surface, making it potentially more subjected to bacterial adhesion and consequently increasing the risk of developing new caries. Furthermore, the reduction in enamel coating may lead to dentine hypersensitivity or periodontal implications, such as alveolar bone loss. However, a metanalysis conducted on different IER methods to evaluate caries incidences after stripping demonstrated that there are no significance differences between treated and untreated teeth surfaces, probably thanks to the physiological conditions of the oral microbiome and to its ability to counterbalance the increased level of enamel roughness due to the stripping procedure [12].

Clearly, the clinician has to make a correct diagnosis in order to evaluate the precise amount of enamel that has to be removed, to avoid over reduction and, consequently, an iatrogenic problem [2]. To help the orthodontists in making this procedure, IER kits usually contain thickness gauges to control the amount of enamel reduction directly on the patients during the stripping procedure. Despite this, as underlined by Jonher et al. [13], there are often differences between the amount of enamel intended to be removed and that actually stripped. It depends on several factors, such as enamel hardness, exerted pressure of the clinician, time used to realize the stripping, and also, type of the abrasive used to make the reduction [14].

To reduce the potential adverse effects of interproximal enamel reduction as much as possible, the most important thing is to polish and finish the surfaces after the stripping procedure [15].

Today, there are several stripping techniques available to clinicians, from the traditional hand-held abrasive strips, tungsten carbide, diamond burs, on a handpiece, to the new powered IER systems, such as oscillating diamond-coated disks or motor-driven abrasive strips, up to the more recent sonic-activated diamond-coated tips [8].

Sheridan in 1985 [11] described a method called Air Rotor Stripping (ARS), that focuses on stripping on posterior teeth, particularly on molars and premolars [16]. Different studies have demonstrated that ARS is useful in the treatment of dental crowding because it generates posterior space, increasing the arch length of approximatively 6.4 mm, but it also increases the susceptibility of interproximal enamel surfaces to demineralization, even if the author proposed to polish and finish treated areas and to apply topical fluoride solutions to prevent secondary lesions.

Some studies have reported that enamel remineralization occurs about one year after the ARS procedure [8,12,17,18], but it is impossible to obtain 100% well-polished surfaces, even after the use of burs and polishing disks after the stripping [7,13,14,19,20]. According to Zachrisson and coauthors [20], very important is the use of adequate water and air cooling while performing stripping, to safely reduce the enamel surfaces.

Different studies, in order to determine the roughness of the enamel before and after the ARS procedure, have analyzed enamel surfaces using 3D profilometric measurements and then confirmed the profilometric results using the SEM images of enamel [21].

The aim of this study was to evaluate the resulting enamel surface conditions after using different methods and devices for interproximal enamel reduction, with different thickness, shapes, and graining size, in order to implement the rare overall data available on this topic.

## 2. Materials and Methods

One-hundred and sixty freshly extracted, caries-free, intact human lower and permanent incisors were included in this study. These teeth were extracted for orthodontic reasons and had been collected in the database of the University of Genoa.

The tooth selection criteria for this study were as follows:Absence of white spots.Absence of caries.Absence of morphological and structural alterations of the interproximal enamel.

All procedures were performed at the Department of Orthodontics of the University of Genoa (Italy). The samples were processed after fixing on plaster support, and then treated with four different techniques of enamel reduction and finishing. The elements were randomly divided into eight groups: A1-A2, B1-B2 C1-C2, D, and E. Each group was composed of 20 teeth, for a total of 160 elements. The A, B, and C groups were divided into two subgroups (A1-A2, B1-B2, and C1-C2) with the following characteristics:Subgroup 1: reduction in the interproximal enamel.Subgroup 2: reduction and finishing.

A1: fine diamond bur (30 microns) Komet 862 EF.

A2: fine diamond bur Komet 862 EF finishing with 12 steps 3M Soft Lex medium–fine–ultrafine disks.

B1: extra-fine diamond bur (15 microns).

B2: extra-fine diamond bur finishing with 12 steps 3M Soft Lex medium–fine–ultrafine disks.

C1: tungsten carbide bur (Komet ET9-8 September 4159).

C2: tungsten carbide bur (Komet ET9-8 September 4159) finishing with 12 steps of medium–fine–ultrafine 3M Soft Lex disks.

D: Horico extra-fine diamond strips (thickness 0.10 mm).

E: controls (untreated enamel).

To obtain comparable operating conditions, the treatments were carried out by a mechanical device operating at a constant pressure and were able to remove from all samples the same enamel thickness (0.5 mm). After performing the different procedures of stripping, the samples were removed from the plaster support. A section along the long axis of the tooth was performed with a diamond blade, separating the mesial surface from the distal one to obtain two fragments for each sample. The fragments were washed with deionized water and dried with an air jet. All samples were mounted on metallic supports (stab), fixed to these by a colloidal graphite (to maintain the electrical conductivity), and metallized with gold for SEM observation (Carl Zeiss EV040 USA). The metallization was carried out to a thickness of 20 nanometers, at a current of 25 mA, and for a time of 1.5 min. The approach used to give an objective evaluation of the performance of the various stripping techniques is based on the application of algorithms to the digital images acquired by SEM The observation of images revealed a main morphological characteristic: the presence of streaks due to the action of the cutters used. To evaluate the presence of streaks, the evaluation of enamel roughness was used (Ra, μm).

Danesh et al. [14] in their article used a profilometric examination of the surfaces treated with IER. Profilometry is a method used to evaluate surface configurations with a noninvasive approach. Photomicrographs of the analyzed surfaces were made before the profilometric examination, then image processing was carried on by a software that allowed a numeric and graphical description of the surfaces. For each surface analyzed, they calculated the arithmetic average value of the roughness of the profile’s deviance from the average (Ra, μm) for the treated surfaces and for the control group.

The evaluation of enamel roughness (Ra, μm) was evaluated usin ImageJ program (https://imagej.nih.gov/ij/, accessed on 12 October 2022) for the SEM image analysis through the 3D view of the peaks and the algorithm for calculating the Ra average value and the standard deviation (IC 95%). In this way, SEM images allowed to visualize grooves, and starting from this average value, a statistical evaluation was then conducted to analyze the overall roughness.

Additionally, Zingler [2] in his article evaluated enamel roughness (Ra, μm) before and after IER and polishing procedures, using a mean value determined after observing teeth surfaces with a confocal laser scanning microscope.

To calculate the standard deviation of the obtained images (A1, A2, B1, B2, C1, C2, and D) a “histogram” function in Adobe Photoshop was used, which calculates the histogram (the distribution of gray levels in picture by a graph, in which the x axis represents the 256 possible values and the y axis includes the number of pixels. The histogram shows for each gray value the number of pixels in the image).

### Statistical Analysis

Median value of Ra (μm) for each group and the 95 confidence interval (interquartile, min–max) were calculated.

The Ra values of the treated groups were compared using parametric t tests with the untreated tooth. The significance threshold was set at 0.05.

## 3. Results

### Enamel Roughness

Comparing the different digital images acquired by SEM, the presence of more or less streaks on IER surfaces after the procedure can be observed.

An evaluation of the enamel roughness (Ra, μm) is shown in Table 1 and in the box plot (Figure 1), and the results indicate that all methods of IER used in this study leave the enamel significantly rougher (*p* < 0.05) compared to the untreated tooth (group E), except for group C2, which leaves the IER surface much more smooth. Tungsten carbide bur (Komet ET9-8 September 4159) is used for the interproximal enamel reduction, and then the finishing is made with twelve steps of 111 medium–fine–ultrafine 3M Soft Lex disks (Figure 2) following this protocol: four steps of 111 medium 3M Soft Lex disk, four steps of 111 fine 3M Soft Lex disk, and finally, four steps of 111 ultrafine 3M Soft Lex disk. For each tooth, a new kit of burr and disks have been used.

## 4. Discussion

Among the different methods used to obtain space in orthodontics, when the therapeutic choice is directed toward making no extractions, one is represented by the expansion of the upper arch, which is perhaps the most frequent orthodontic therapies in growing patients [22,23,24,25], and is also often necessary to avoid other orthodontics and functional problems. Another treatment option is represented by interdental stripping or interproximal enamel reduction, a commonly applied technique used to obtain space to align incisors and maintain alignment in the long term [11], as indicated in a recent systematic review [6]. However, this method has a downside: the removal of part of the enamel surface can make the patient more susceptible to interproximal demineralization, caries, or periodontal lesions.

In this study, the objective was to analyze the surfaces enamel conditions after interproximal enamel reduction. Among the different and widely accepted methods, SEM images were taken to visualize grooves and trenches, but the observer’s subjectivity represented a limitation to this analysis. So, the approach used to give an objective evaluation of the performance of the various stripping techniques was based on the application of the digital image processing algorithms acquired by SEM. The observation of these images at the end of the stripping procedure revealed a main morphological characteristic, namely, the presence of streaks due to the action of the cutters used.

Our study demonstrates that all the stripping methods used have left strikes on enamel surfaces, but that the use of the milling cutter of tungsten carbide Komet ET9-8 Set 4159 followed by a finishing procedure with 12 steps of a 111 medium, fine, and ultrafine 3M Soft Lex disks, has guaranteed the least roughness of the treated surfaces.

Certainly, as it is also mentioned by Lombardo et al. [15], the finer the grain size of the burr or the disk used for the enamel reduction, the easier the polishing procedures, which are necessary to refine teeth surfaces. In their study, the authors for the first time realized a stripping protocol, underlying the importance of an adequate polish of the stripped surfaces after the IER procedure, much like other authors have previously suggested.

Lundgren et al. [26] in their study used a contact profilometer to evaluate the surface texture after stripping, and 2D parameters were compared. The research performed with SEM proved that the stripped surfaces had undergone some serious changes and that there was a broad range of the mean values of the 2D parameters, and so, the conclusion was that it was impossible to eliminate furrows produced by ARS [27] with diamond burrs, diamond disks, and 16-blade tungsten carbide burs used in the normal procedures of polishing and cleaning. This result is confirmed by the study of Arman [8], where the SEM analysis showed the presence of grooves in enamel-treated surfaces. In his study, Arman et al. have also evaluated how enamel microhardness is influenced from the stripping technique. After the analysis, they confirmed that the different methods analyzed did not modify the mineral content of the enamel of the surfaces treated. Otherwise, deciduous tooth enamel showed up more porous than permanent enamel, and consequently, it was less hard and more susceptible to erosion after stripping.

Piacentini and Sfondrini [19] recommended the use of Soft-Lex disks after enamel reduction with tungsten-carbide burs and diamond-coated disks, respectively. The crucial factor when stripping teeth is the last IER working end used, which must have a very fine grit, but the authors underlined that the conventional methods of polishing are insufficient to remove the trenches left on enamel surfaces after grinding them with burs. Furthermore, ideally a new kit of disks should be used for each interproximal surface to guarantee an ideal finishing. In the same way, Zingler et al. [2] underlined how roughness values were significantly higher before polishing, which became necessary after all the stripping procedures.

According to Zhong et al. [7], more than 90% of the stripped enamel surfaces were well-polished and seemed to be smoother and less plaque-retentive than the untreated enamel surfaces, but they also draw attention to the cariogenic implication that an uncareful application of ARS may lead to. This corresponds to the results of the study by Mikulewicz [16].

Among the different procedures analyzed until now, Joseph et al. [28] proposed chemical stripping, resulting in an etched adamantine surface that could become susceptible to rapid plaque accumulation and consequent demineralization. The authors demonstrated that the use of Soft Lex polishing disks after interproximal enamel reduction allows the acquisition of smoother enamel surfaces; however, emphasizing that this procedure does not help to achieve the same morphology characteristics as with intact enamel.

Johner et al. [13], using a three-dimensional laser scanner, investigated the predictability of the expected stripping amount on premolars using three different stripping device. They concluded that, in most cases, actual stripping was less than the intended amount of enamel reduction and that the stripping technique did not appear to be a significantly predictive element of the actual amount of enamel reduction performed.

The systematic review by Koretsi [12] reported that researchers mainly employed scanning electron microscopy (SEM) to investigate the enamel roughness that results from stripping. Although SEM is an excellent means of visualizing the topographical characteristics of a surface in detail, it does not allow for a comparison of the findings owing to the absence of a quantitative scale that provides objectivity and reproducibility of the measurements. Furthermore, the variability inter- and intra-observer can be high, which means that the results need to be interpreted carefully.

Most studies that examined the surface characteristics of the enamel ground by stripping procedures were limited to qualitative evaluations with SEM. Microscopic analyses can provide only a visualization of surface morphology, necessitating the use of further methods to quantitatively determine the extent of surface modifications. In this study, we used digital algorithms applied to SEM images to give an objective evaluation of the performance of the various stripping techniques. The use of a mechanical device for enamel reduction and a digital analysis of the images by means of automatic algorithms made it possible to have accurate and objective information on the performance of the different stripping techniques.

In recent years, interproximal enamel reduction has become a widespread clinical procedure that is indicated to resolve different orthodontic problems not only in adults, but also in adolescents and in children. Despite the iatrogenic effects that IER can determine, nowadays it is clear that if performed in the correct way and with adequate precautions, all these collateral effects can be avoided. For this reason, the patient’s motivation are fundamental. The clinician has to educate patients about the adequate hygiene instructions, including interproximal plaque control after the procedure of interdental enamel reduction, in order to avoid negative consequences due to the accumulation of plaque in the treated areas.

In this study, there are some limitations. First of all, the procedure is conducted on healthy teeth, so the presence of carious lesions, abrasions, or other periodontal problems could determine a different reduction in the enamel surface at the end of the procedure. Mesio-distal diameters of each lower incisor were not measured at the beginning of the study, so the reduction created with the IER procedure could have not been the same for all teeth, and the starting age of the teeth was not known. It is known that the amount of enamel in each tooth is different depending on a person’s age; moreover, the procedure followed in our study did not include the application of fluorine varnishes, which should have remineralized the enamel surfaces after IER, even if recent studies have indicated that this procedure is unnecessary [15].

These elements could have represented problems, but the simple size of the study and the randomized distribution in groups should have reduced these possible biases.

Surely, future studies that are focused on the predictability of interdental enamel reduction should be conducted in a clinical setting, where the reproximated surfaces are constantly remineralized from the exposure to saliva and fluoride toothpaste, which can also evaluate the patient’s comfort and the ease of execution for the clinician, with the use of different stripping devices.

## 5. Conclusions

All the different methods of IER leave streaks on the enamel surfaces. The stripping technique that allows the removal of interproximal dental tissue, guaranteeing the least roughness of the treated surface and then the lower morphological alteration of enamel surfaces, is that which involves the use of the milling cutter of tungsten carbide Komet ET9-8 Set 4159, finishing with 12 steps of 111 medium, fine, and ultrafine 3M Soft Lex disks with the following procedure: four steps of 111 medium 3D Soft Lex disk, four steps of 111 fine 3D Soft Lex disk, and finally, four steps of 111 3D Soft Lex disk.

## Figures and Tables

**Figure 1 jfb-14-00110-f001:**
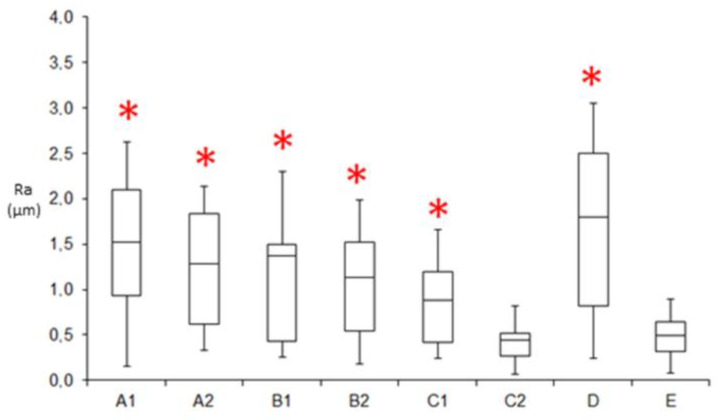
Box plot of enamel roughness (Ra, µm). The asterisk symbol (*) indicates that all the groups analyzed have significant values of enamel roughness (Ra > 0.05) compared to the control group (E), except for group C2.

**Figure 2 jfb-14-00110-f002:**
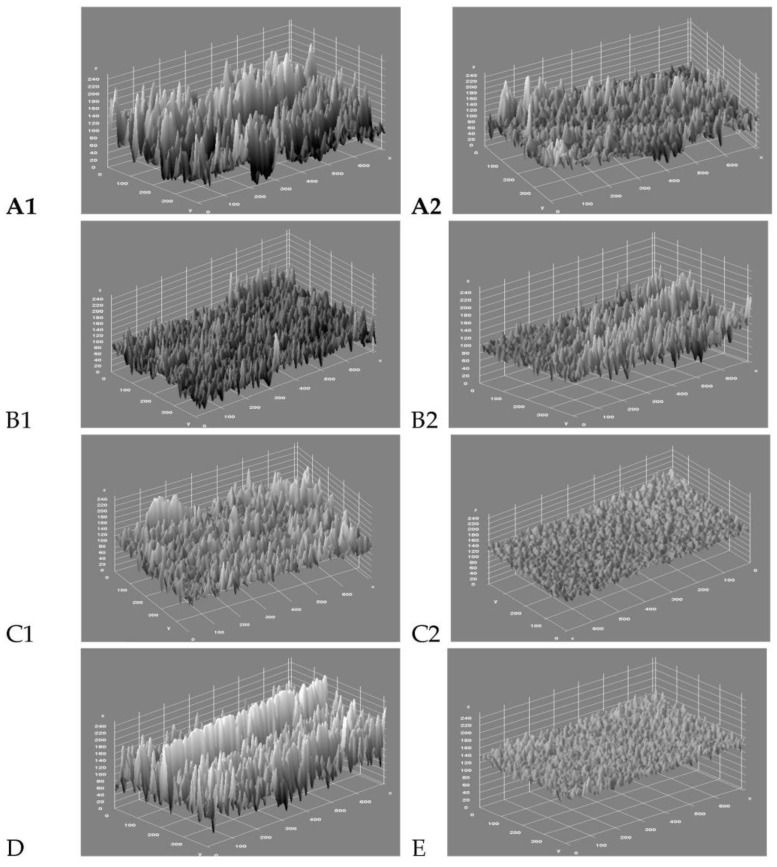
Evaluation of enamel roughness (Ra, µm) with ImageJ program. (**A1**): fine diamond bur (30 microns) Komet 862 EF. (**A2**): fine diamond bur Komet 862 EF finishing with 12 steps 3M Soft Lex medium–fine–ultrafine disks. (**B1**): extra-fine diamond bur (15 microns). (**B2**): extra-fine diamond bur finishing with 12 steps 3M Soft Lex medium–fine–ultrafine disks. (**C1**): tungsten carbide bur (Komet ET9-8 September 4159). (**C2**): tungsten carbide bur (Komet ET9-8 September 4159) finishing with 12 steps of medium–fine–ultrafine 3M Soft Lex disks. (**D**): Horico extra-fine diamond strips (thickness 0.10 mm). (**E**): controls (untreated enamel).

**Table 1 jfb-14-00110-t001:** Enamel roughness (Ra, μm) values for each group.

	A1	A2	B1	B2	C1	C2	D	E
Q_1_	0.934	0.623	0.434	0.543	0.423	0.261	0.824	0.323
Median	1.516	1.281	1.374	1.135	0.876	0.440	1.797	0.489
Q_3_	2.100	1.835	1.500	1.524	1.2	0.518	2.500	0.647
Max	2.619	2.135	2.293	1.991	1.652	0.818	3.048	0.898
IQR	1.166	1.211	1.066	0.981	0.777	0.256	1.676	0.324
Min	0.153	0.327	0.255	0.180	0.240	0.061	0.247	0.080

## Data Availability

The data presented in this study are available on request from the corresponding author.

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
