# Peer review of "Evaluation of Enamel Surfaces after Different Techniques of Interproximal Enamel Reduction"

_jfb, 2023, doi:10.3390/jfb14020110_

Round 1

Reviewer 1 Report

Dear authors,

You all did excellent work. Your article offers very insightful information on the techniques for interproximal enamel reduction.

You chose an interesting and challenging topic to discuss in your paper, which was very well-argued.

This excellent paper provided relevant advantages of using the stripping techniques in the removal of interproximal enamel.

Keep up the excellent work!

Author Response

Dear authors,

You all did excellent work. Your article offers very insightful information on the techniques for interproximal enamel reduction.

You chose an interesting and challenging topic to discuss in your paper, which was very well-argued.

This excellent paper provided relevant advantages of using the stripping techniques in the removal of interproximal enamel.

Keep up the excellent work!

Thank you for your valuable opinion and encouragement

Reviewer 2 Report

Manuscript pdf sent with due corrections.

Author Response

Ra units are missing

Ra units has been added

this row is repeated

Table has been fixed

The legend of the figures appears at the bottom and not at the top.  Proceed in the same way for the other figures.

The legends have been fixed

SEM micrographs (B1-B2, C1-C2, D and E) are not scaled. The scale is fundamental in all micrographs obtained by SEM as well as the definition of the parameters used.

Thanks for reporting. The scale is not shown on all the high definition stills in our possession. Some have low-definition, not suitable for publications. Evaluations with ImageJ software were all done in the range 500x-524x magnification. Below I attach the photographs in the same magnification scale.

A1          

A2          

B1          

B2          

C1           

C2      

 D           

 E          

in my opinion, the authors are comparing incomparable things. the micrographs have completely different magnifications, so the method cannot be used clearly.

Furthermore, in a scientific work in which the intention is to characterize the surface, to determine only Ra as roughness parameters is clearly insufficient. How about Rq, Rz, Rp, Rv. Rsk, Rku, ???

Thanks for raising the issue. Following a re-evaluation based on your suggestion, we decided to drop the Hought transform to avoid misunderstandings.

Regarding the decision to use only RA, this choice was made for this first clinical pilot work. We believe that RA gives a clear and simple initial assessment that can be immediately helpful and understandable to the clinician. Furthermore, we are already working to increase the sample and the variables considered, as suggested by the reviewer.

Reviewer 3 Report

Thank you for this research which can contribute to making the decision about the procedure that clinicians going to perform. 

The manuscript is missing some information in order to be easily understood and read:

- in materials and methods it is not clear when the selection of the teet (in groups ) was performed, I guess before the procedure, but please make this clear. 

- add limitations of the study and number all that could be for example MD diameter of the tooth was not measured before the study. Lower incisors have different mes- distal diameters, so no 0.5 reduction is not the quite same for all teeth.

-add limitation regarding the possible age as this could also contibute to results of the study

- add limitation regarding no fluoride use as the correct procedure is that you always perform fluoride varnish after enamel reduction

- add limitation regarding the burs - are the new ones is always used or not

In discussion row 251 please add that these are the options only if non-extraction treatment is to be planned 

In the discussion add the main results of your study and make a comparison with other studies you already mentioned. 

Update reference list - too many old references

Author Response

Thank you for this research which can contribute to making the decision about the procedure that clinicians going to perform. 

The manuscript is missing some information in order to be easily understood and read:

- in materials and methods it is not clear when the selection of the teet (in groups ) was performed, I guess before the procedure, but please make this clear. 

One hundred and sixty lower incisors have been extracted and collected in the database of University of Genoa and then divided randomly in eight groups before the procedure. The teeth used for the study belonged to patients requiring extraction of a lower incisors for orthodontics reasons.

I have followed this indication and explained it better in the text.

- add limitations of the study and number all that could be for example MD diameter of the tooth was not measured before the study. Lower incisors have different mes- distal diameters, so no 0.5 reduction is not the quite same for all teeth.

-add limitation regarding the possible age as this could also contibute to results of the study

- add limitation regarding no fluoride use as the correct procedure is that you always perform fluoride varnish after enamel reduction

The study has some limits. Mesio-distal diameters of each lower incisor were not measured at the beginning of the study, so the reduction created with IER procedure could be not the same for all teeth, and the age of the teeth is not known; moreover, the procedure did not include the application of fluorine varnishes which should remineralize the enamel surfaces after IER, even if recent findings  seem to imply that this procedure might be unnecessary (Lombardo et al.). These elements can represent problems, but the simple size of the study and the randomized distribution in groups reduce this possible bias.

I have added a paragraph with the limits of the study.

- add limitation regarding the burs - are the new ones is always used or not

For each group each new bur was used in the same number of teeth.

In discussion row 251 please add that these are the options only if non-extraction treatment is to be planned 

Done, as requested

In the discussion add the main results of your study and make a comparison with other studies you already mentioned. 

Yes, I’ve added the results of our study in the discussion and compared them with other studies.

Update reference list - too many old references

I’ve updated reference list.

Reviewer 4 Report

Dear Author, congratulating you on your work I recommend:

- Declare how the sample used was determined.

- Explain the final polishing protocol (12 steps of 111 medium-fine-ultrafine 3M Soft Lex discs)

- The name of the image program, the final j is with a capital letter (ImageJ).

- Table 1 should go in the results section, not in the methodology.

Best regards

Author Response

Dear Author, congratulating you on your work I recommend:

- Declare how the sample used was determined.

One hundred and sixty lower incisors have been extracted and collected in the database of University of Genoa and then divided randomly before the procedure in eight groups. The teeth used for the study belonged to patients requiring extraction of a lower incisors for orthodontics reasons.

- Explain the final polishing protocol (12 steps of 111 medium-fine-ultrafine 3M Soft Lex discs)

The final polishing protocol after IER involves four steps of 111 medium 3M Soft Lex disc, four steps of 111 fine 3M Soft Lex disc and finally other four steps of 111 ultrafine 3M Soft Lex disc.

The sentence has been added.

- The name of the image program, the final j is with a capital letter (ImageJ).

Ok thanks, I’ve written the name of the image program wrong: ImageJ. I’ve corrected it in the article.

- Table 1 should go in the results section, not in the methodology.

Ok thanks, I moved the table in the result section.

Round 2

Reviewer 2 Report

The paper can be accepted for publication